# Body Composition Analysis in Metastatic Non-Small-Cell Lung Cancer: Depicting Sarcopenia in Portuguese Tertiary Care

**DOI:** 10.3390/cancers17030539

**Published:** 2025-02-05

**Authors:** José Leão Mendes, Rita Quaresma Ferreira, Inês Mata, João Vasco Barreira, Ysel Chiara Rodrigues, David Silva Dias, Manuel Luís Capelas, Antti Mäkitie, Inês Guerreiro, Nuno M. Pimenta, Paula Ravasco

**Affiliations:** 1Medical Oncology Department, Unidade Local de Saúde São José, 1169-050 Lisbon, Portugal; ritaquaferreira@hotmail.com (R.Q.F.); ines.m.guerreiro@gmail.com (I.G.); 2Centro Clínico Académico de Lisboa, 1169-056 Lisbon, Portugal; inesgm4@gmail.com; 3Center for Interdisciplinary Research in Health (CIIS), Universidade Católica Portuguesa, 1649-023 Lisbon, Portugal; daviddias_77@hotmail.com (D.S.D.); luis.capelas@ucp.pt (M.L.C.); antti.makitie@helsinki.fi (A.M.); npimenta@esdrm.ipsantarem.pt (N.M.P.); pravasco@ucp.pt (P.R.); 4Radiology Department, Unidade Local de Saúde São José, 1169-050 Lisbon, Portugal; 5Medical Oncology Department, CUF Oncologia, 1998-018 Lisbon, Portugal; joaovascobarreira@gmail.com; 6Faculty of Health Sciences and Nursing, Universidade Católica Portuguesa, 1649-023 Lisbon, Portugal; 7Medical Oncology Department, Fundação Champalimaud, 1400-038 Lisbon, Portugal; y.chiara@gmail.com; 8Medical Oncology Department, Unidade Local de Saúde Cova da Beira, 6200-251 Covilhã, Portugal; 9Faculty of Health Sciences, Universidade da Beira Interior, 6200-251 Covilhã, Portugal; 10Department of Otorhinolaryngology—Head and Neck Surgery, Helsinki University Hospital, 00014 Helsinki, Finland; 11Research Program in Systems Oncology, Faculty of Medicine, University of Helsinki, 00014 Helsinki, Finland; 12Faculty of Medicine, Universidade Católica Portuguesa, 2635-631 Rio de Mouro, Portugal; 13Rio Maior School of Sport, Santarém Polytechnic University, 2040-413 Rio Maior, Portugal; 14Sport Physical Activity and Health Research and Innovation Center (SPRINT), 2040-413 Rio Maior, Portugal; 15Egas Moniz Center for Interdisciplinary Research, Instituto Universitário Egas Moniz, 2829-511 Almada, Portugal

**Keywords:** non-small-cell lung carcinoma, sarcopenia, body composition, skeletal muscle index, prognosis

## Abstract

Sarcopenia (low skeletal muscle mass) is an emerging prognostic factor in cancer, but standard definitions overlook heterogeneous muscularity across populations. This study evaluated the prognostic value of a cohort-specific sarcopenia definition compared to a standard literature definition in Portuguese patients with metastatic non-small-cell lung cancer (mNSCLC) receiving first-line palliative treatment. Among 184 patients, 66.3% were classified as sarcopenic by the literature definition, whereas 46.7% were identified using the cohort-specific definition. The cohort-specific definition better predicted shorter overall survival (12.75 vs. 21.13 months) and progression-free survival (7.92 vs. 9.56 months). In addition, sarcopenic obesity was linked to worse outcomes, while being overweight (but not obese) improved survival among sarcopenic patients. These findings highlight the need for tailored strategies to address muscle loss in cancer care.

## 1. Introduction

Lung cancer (LC) is the leading cause of cancer morbidity and mortality both worldwide and in all European countries. Male-to-female incidence and mortality ratios range from one to five-fold [1]. Tobacco remains the main risk factor: in developed countries, smoking trends among women hint at LC incidence nearing that in men; however, in lower-income countries, smoking rates are still peaking among men [1,2,3]. Low-dose CT screening has shown to reduce LC mortality; although, false positive rates, overdiagnosis and biopsy complications are of concern. Currently, screening in Europe essentially relies on pilot programs [1,4,5].

Non-small-cell LC (NSCLC) comprises 85% of all LCs. Adenocarcinoma (50–60%) and squamous-cell carcinoma (20–30%) are predominant [6]. Up to 60% of lung adenocarcinomas are reported to harbor a driver mutation, depending on world region and smoking history [7,8]. In all patients with advanced NSCLC and unusual lung squamous-cell carcinomas, it is recommended to perform standard genome sequencing including KRAS, EGFR, ALK, ROS1, NTRK, RET, MET, BRAF and HER2. When present, actionable mutations require tailored treatment [7]. Programmed death-ligand 1 (PD-L1) tumor proportion scoring (TPS) is mandatory given it predicts immunotherapy (IO) efficacy [9].

Although we navigate in the precision oncology era, body composition information remains neglected regarding treatment decision. Body surface area (BSA) (e.g., DuBois), used in cancer treatment dose scaling, fails to discern body composition. Indeed, skeletal muscle (SM) does not correlate with BSA in cancer patients [10]. Sarcopenia comprises both the loss of muscle mass and function (i.e., strength) negatively impacting health [11]. Of note, the recent focus on SM depletion, whether rooted in toxicity prediction or prognostic value, has outpaced research focusing on muscle strength as far as clinical oncology is concerned [10,12]. Publications from Baracos et al., favoring the exploitation of standard CT-scan imaging for body composition analysis, as well as from Wei Shen et al., demonstrating a high correlation between whole body muscle mass and cross-sectional SM area (SMA) at the third lumbar vertebra (L3), have shaped the current framework for defining sarcopenia in cancer patients: optimal thresholding SMA normalized for stature, i.e., an SM Index (SMI) [13,14].

Notwithstanding, proposed cut-offs in the literature are heterogeneous [10,12]. Caucasian-predominant SMI thresholds for mortality, as published by Prado et al. [15] (later extended by Martin et al. to include non-obese patients [16]) or Fearon et al. [17], are discrepant to Asian-specific published thresholds [10,18,19,20]. Such discrepancy not only highlights caveats in how these definitions translate to different ethnicities but foresees shortcomings when applying them to cohorts of mixed cancers or cancer stages.

Sarcopenia has been shown to impact survival in various cancers, including NSCLC [20,21]. Most studies thresholding L3SMI for prognosis in NSCLC are Asian and lack homogeneity regarding both cancer stage, cancer treatment and treatment setting [20]. Of note, evidence on EGFR-mutant NSCLC remains mixed [20,22,23].

Unstandardized sarcopenia definitions preclude timely multimodal interventions for reversing muscle loss and performance status (PS) optimization that enable standard of care oncological treatment. This study assesses the impact of SMI optimal thresholding on sarcopenia rates and prognosis within a Portuguese metastatic NSCLC (mNSCLC) population. In addition, we discuss the discrepancies within the thresholds presented in the literature.

## 2. Materials and Methods

### 2.1. Procedures

This is a retrospective analysis on data collected from patients with mNSCLC treated at the Unidade Local de Saúde São José (ULSSJ) Medical Oncology Department between January 2017 and December 2022. We collected data on patient variables (sex, age at NSCLC diagnosis, smoking status, Eastern Cooperative Oncology Group [ECOG] PS and anthropometric data: height and weight at the start of systemic treatment in metastatic setting, first-line [1 L]), cancer variables (American Joint Committee on Cancer [AJCC] staging version 8, NSCLC subtype, mutational status, PD-L1 TPS and metastatic sites) and treatment variables (treatment protocols and response assessment imaging). Cross-sectional CT-scan images at L3 level starting 1L treatment in metastatic setting were analyzed using the National Institute of Health ImageJ v1.54 g software [24]. Wacom One was used to calculate SMA (https://www.wacom.com/en-us/products/pen-tablets/one-by-wacom (accessed on 30 June 2024)), including psoas major, quadratus lumborum, erector spinae, latissimus dorsi, abdominal oblique muscles and rectus abdominis. SMA was measured in square centimeters (cm^2^) using a Hounsfield Unit (HU) range of −29–150 HU. SMI was calculated by dividing the SMA by height squared (cm^2^/m^2^).

The study was conducted according to the Declaration of Helsinki and was approved by the Ethics Committee for Health of the ULSSJ in 2024 with a waiver for informed consent.

### 2.2. Patients

The study population was identified through screening the ULSSJ Pathology Department’s files for histological diagnoses coded by the clinical terms in the Systemized Nomenclature of Medicine (SNOMED)/International Classification of Diseases for Oncology (ICDO) as “lung” (T-28000.01/T.C34.9), “adenocarcinoma” (M-81403.01/M.8140.3-G), “squamous-cell” (M-80703.01/M.8070.3-G), “adenosquamous” (M-85603.01/ M.8095.3-G) and “carcinoma, NOS” (M-80103.01/ M.8010.3-G). Duplicates were excluded, and the following exclusion criteria were applied: <18 years old; no records of Medical Oncology outpatient clinic; no primary LC (i.e., SNOMED/ICDO corresponding to secondary LC/lung metastasis of primary tumor with different origin); neuroendocrine LC (large/small-cell); adenoid cystic carcinoma; carcinoid tumor; thymic cancer; AJCC stage III LC without progression after chemoradiotherapy (irrespective of IO consolidation treatment); AJCC stage IV LC not progressing after treatment with radical intent; patients that did not receive oncological treatment (i.e., exclusive Best Supportive Care); and patients with synchronous malignancies, except for basal cell carcinomas.

### 2.3. Definitions and Endpoints

The primary endpoint was overall survival (OS), defined as time from mNSCLC diagnosis to death from any cause. The secondary endpoint 1L progression-free survival (PFS) was defined as time from starting 1L treatment in a metastatic setting until disease progression or death from any cause. Two sarcopenia definitions—as published by Prado et al. (SMI < 52.4 cm^2^/m^2^ for men and <38.5 cm^2^/m^2^ for women) [15] and defined using SMI cohort-specific cut-offs)—were applied to statistical analysis based on relevance, i.e., both definitions were conditional to accurate survival stratification. Obesity was defined according to the World Health Organization definition (body mass index [BMI] ≥ 30 kg/m^2^). Sarcopenic obesity was defined as simultaneous obesity (i.e., BMI ≥ 30 kg/m^2^) and sarcopenia (as published by Prado et al. [15] or as defined within the study population). Other secondary endpoints included both 1L treatment and BMI subgroup analyses regarding OS. The follow-up data cut-off was 15 July 2024.

### 2.4. Statistical Analysis

Statistical analysis was performed using SPSS version 25. A two-tailed *p*-value of 0.05 was considered statistically significant for all performed tests. Continuous variables were reported as means and their standard deviations. Comparisons between categorical variables were assessed using Chi-square tests. Optimal SMI thresholding was obtained by receiver operating characteristic analyses. The Kaplan–Meier method and log-rank tests were used for survival analyses. A multivariate cox regression model was performed, including variables showing univariate association with OS. Missing data were handled based on the listwise deletion method.

## 3. Results

One hundred ninety-seven patients with mNSCLC met the prespecified inclusion criteria. The mean age was 65 years (standard deviation ± 11.31). Most patients were male (*n* = 135), with reported former/active smoking habits (*n* = 103). Adenocarcinomas were predominant (*n* = 165), and most tumors were metastatic at presentation (*n* = 154). Baseline characteristics of the initial cohort are shown in Table 1. SMI was evaluable in 184 patients: mean SMI was 48.52 cm^2^/m^2^ (±9.31) for men and 37.69 cm^2^/m^2^ (±6.14) for women. Nutritional data collection and endpoint testing analysis were limited to this cohort. Body composition data of the SMI-assessed cohort are shown in Table 2. Optimal sex-specific SMI thresholds were <49.96 cm^2^/m^2^ for men and <34.02 cm^2^/m^2^ for women. A total of 122 patients (66.30%), corresponding to 89/125 men (71.20%) and 33/59 women (55.93%) were sarcopenic as defined per Prado et al., whereas 86 patients (46.74%), corresponding to 73/125 men (58.40%) and 13/59 women (22.03%) were sarcopenic as defined per optimal SMI thresholding: 36 patients (19.57%) were reclassified as not sarcopenic. After reclassification, 14/86 sarcopenic patients were underweight (BMI < 18.5 kg/m^2^), 17 were overweight but not obese (BMI ≥ 25 and <30 kg/m^2^) and 5 had sarcopenic obesity (BMI ≥ 30 kg/m^2^), while the remnant 50 had normal weight (BMI ≥ 18.5 kg/m^2^ and <25 kg/m^2^).

At the data cut-off, 18 out of 184 patients remained alive without progression, while 34 out of 184 patients remained alive, with or without progression. Median PFS was 8.91 months (95% confidence interval [CI] 7.46–10.35). Prado et al.’s definition for sarcopenia did not predict PFS in our cohort (8.87 months vs 8.91 months; *p* = 0.392), which is contrary to the cohort-specific thresholds (7.92 months vs 9.56 months; hazard ratio [HR] 1.503; 95% CI 1.1–2.05; *p* = 0.01). Median OS was 18.4 months (95% CI 14.79–22.01). Prado et al.’s definition did not predict OS (17.9 months vs 20.11 months, *p* = 0.588). Conversely, the cohort-specific sarcopenia thresholds were prognostic (12.75 months vs 21.13 months; HR 1.654; 95% CI 1.20–2.29; *p* = 0.002). Amid sarcopenia, patients presenting a BMI ≥ 25 kg/m^2^ were at a lesser risk of death (HR 1.084; 95% CI 0.069–1.927) when compared to patients with a BMI < 25 kg/m^2^ (HR 1.904; 95% CI 1.231–2.944). Sarcopenia’s survival impact was consistent across 1L treatment subgroups. The Kaplan–Meier plots for OS, as well as the between-group difference in OS (HR for death) for sarcopenic patients (defined per cohort-specific thresholds), are illustrated in Figure 1 and Figure 2, respectively.

BMI significantly stratified survival among sarcopenic patients (*p* = 0.002). Median OS were as follows: underweight (10.22 months; 95% CI 5.526–14.914), normal weight (9.1 months; 95% CI 3.753–14.447), overweight (32 months; 95% CI 15.552–48.448), obese (6.14 months; 95% CI 0.0–13.118). Bearing normal weight as reference group, being underweight was not prognostic (HR 1.378; *p* = 0.321), being overweight decreased mortality (HR 0.417; *p* = 0.01) and obesity increased mortality (HR 2.723; *p* = 0.039). The Kaplan–Meier plot for OS in sarcopenic patients according to BMI is shown in Figure 3. Obesity reduced the risk for sarcopenia (odds ratio 0.34; *p* = 0.039). In multivariate analysis, sarcopenia, underweight and ECOG PS (0 vs ≥1) remained prognostic (shown in Table 3).

## 4. Discussion

The mounting literature linking sarcopenia with survival among various cancer types, stages and treatment settings cement it as an emergent key prognostic biomarker in cancer patients. Regulatory functions concerning insulin-dependent glucose uptake or interactions between myokines and organs such as the liver or brain provide a rationale for this association [25].

Delving into NSCLC, namely studies thresholding L3SMI for prognosis, the literature on the topic is vast. Kimura et al. reported an SMI < 41 cm^2^/m^2^ for men and <38 cm^2^/m^2^ for women as prognostic in a Japanese advanced NSCLC cohort (88.1% stage IV) receiving chemotherapy or EGFR-tyrosine kinase inhibitors (TKI), yielding 38.3% sarcopenic patients [26]. Two Japanese studies thresholding L3SMI at <43.75 cm^2^/m^2^ for men and <41.1 cm^2^/m^2^ for women were also prognostic among stage I NSCLC patients, with sarcopenia rates ranging between 38.8 and 42.2% [27,28]. In another study, Kim et al. reported a 22.4% sarcopenia rate among a Korean NSCLC cohort in pre-operative setting; albeit, sarcopenia, as defined per Fearon et al. (i.e., SMI at <55 cm^2^/m^2^ for men and <39 cm^2^/m^2^ for women), was not prognostic [29]. Likewise, a Croatian study thresholding SMI as per Fearon et al. in advanced NSCLC cohorts, which reported 47% sarcopenic patients, could not predict mortality in patients treated with chemotherapy [30]. Lastly, Stene et al. did not find sarcopenia as defined per Prado et al. to be prognostic within a Norwegian advanced NSCLC cohort treated with chemotherapy and with a sarcopenia rate of 74% [31]. In the study herein, cohort-specific (<49.96 cm^2^/m^2^ for men and <34.02 cm^2^/m^2^ for women) and Prado et al.’s SMI thresholds were compared for prognostic analysis in an mNSCLC cohort. Amid reclassification of 36 (19.56%) patients, sarcopenia rates were 66.30% and 46.74% according to Prado et al. and the cohort-specific thresholds, respectively; notwithstanding these being retrospectively assessed thresholds, such discrepancy is key to understand how the literature definition often cannot accurately translate to different settings.

More recently, similar design studies delved into NSCLC treated with immunotherapy. In two Chinese studies thresholding L3SMI as per Martin et al. (i.e., SMI < 43 cm^2^/m^2^ in men with BMI < 25 kg/m^2^ or <53 cm^2^/m^2^ if BMI > 25 kg/m^2^ and SMI < 41 cm^2^/m^2^ for women irrespective of BMI) [16], in advanced NSCLC cohorts treated with first and second-line immunotherapy, sarcopenia was prognostic regarding overall and progression-free survival [32,33]. Conversely, sarcopenia as defined per Fearon et al. could not predict mortality in an Italian cohort [34]. Noteworthy, Bolte et al. were successful analyzing a 92-patient cohort treated with 1L chemoimmunotherapy, defining sarcopenia based on the psoas muscle index 25th percentile [35]. Sarcopenia rates among these studies ranged between 26–68.9% [32,33,34,35].

Regarding oncogene-addicted NSCLC, two studies focusing on EGFR-mutant cohorts found sarcopenia defined as per Fearon et al. prognostic, with sarcopenia ranging between 54–60.6% [23,36]. Contrariwise, Wu et al. evaluated 176 advanced NSCLC patients treated with 1L afatinib, yielding 53.41% sarcopenic patients; L3SMI as per Prado et al. was not prognostic [22].

Ultimately, the reported studies, consistent regarding cancer type (NSCLC), highlight limitations inherent to broadly applying the L3SMI literature definitions. Of note, the available literature does not suggest that sarcopenia could hold a heterogeneous prognostic value depending on the chosen systemic treatment. Our study, although unbalanced concerning treatment subgroups, aligns with the same proposition (HR for death 1.615, 1.422 and 1.393 for TKI, immunotherapy and chemotherapy subgroups, respectively; shown in Figure 2). Our study is less informative with regards to chemoimmunotherapy, since less than 5% (*n* = 9) of these patients comprised the SMI-analyzed cohort. Notwithstanding, immunotherapy-treated patients were not discrepant to chemotherapy-treated patients within subgroup analysis. Moreover, including chemoimmunotherapy-treated patients in the chemotherapy cohort did not meaningfully change the HR for death or the respective confidence interval (HR for death 1.393 and 1.386 and 95% CI 0.943–2.057 and 0.941–2.041 for chemotherapy and chemotherapy/chemoimmunotherapy, respectively; shown in Figure 2).

BMI stratified OS among sarcopenic patients. Remarkably, HR for mortality resembled the ‘U-shape’ curve described in the context of the obesity paradox [37]. Sarcopenic obesity represented a particularly poor prognostic subgroup: well-known theses on this issue include a greater risk for both cardiovascular disease and mortality, as well as conventional BSA-adjusted dose scaling possibly disproportioning the absolute treatment dose to distribution volume ratio, hence increasing iatrogenesis and mortality. Conversely, being overweight without progressing to obesity may offer metabolic advantages facing a consumptive syndrome [10,38].

To the best of our knowledge, this is the first Portuguese study thresholding L3SMI for prognosis in mNSCLC. In a similar fashion, sex-specific L3SMI cut-off at <49.12 cm^2^/m^2^ and <35.85 cm^2^/m^2^ for men and women, respectively, predicted both mortality and dose-limiting toxicities in a Portuguese metastatic colorectal cancer cohort [39]. The proximity between such cut-offs and those reported herein favors the hypothesis that ethnicity/world-region and treatment setting can be pivotal to define L3SMI-based sarcopenia in cancer patients.

Our study has several limitations. Firstly, its retrospective, single-center design, as well as exploiting imaging not primarily intended for research. Concerning treatment subgroups, 1L chemoimmunotherapy, current standard of care for most mNSCLC patients without contraindications [7,9], was not reimbursed by the Portuguese national health system until May 2022, resulting in less patients receiving this treatment. In addition, providing the hypothesis that sarcopenia could hold different prognostic values dependent on the given systemic treatment, this study disregards the impact of second and subsequent treatment lines on prognosis. Importantly, the study lacks an independent validation cohort. Rather than broadly define L3SMI thresholds for sarcopenia, the study aims to shed light on critical challenges which hinder the sarcopenia definition in clinical practice, while providing a foundation for prospective investigation.

## 5. Conclusions

Cohort-specific L3SMI thresholds independently predicted worse OS and PFS in mNSCLC patients receiving 1L treatment, whereas sarcopenia defined by Prado et al. lacked prognostic significance. Albeit a heterogeneous treatment landscape, sarcopenia’s prognostic value remained consistent across treatment subgroups. On the other hand, BMI further stratified survival outcomes: while being overweight increased survival among sarcopenic patients, sarcopenic obesity increased the risk for mortality. Future studies should prospectively assess tailored approaches to defining L3SMI thresholds in cancer patients, namely with regards to treatment setting and ethnicity, while simultaneously acknowledging the interplay between BMI and skeletal muscle mass. CT-derived sarcopenia analysis is systematically applicable to routine cancer care.

## Figures and Tables

**Figure 1 cancers-17-00539-f001:**
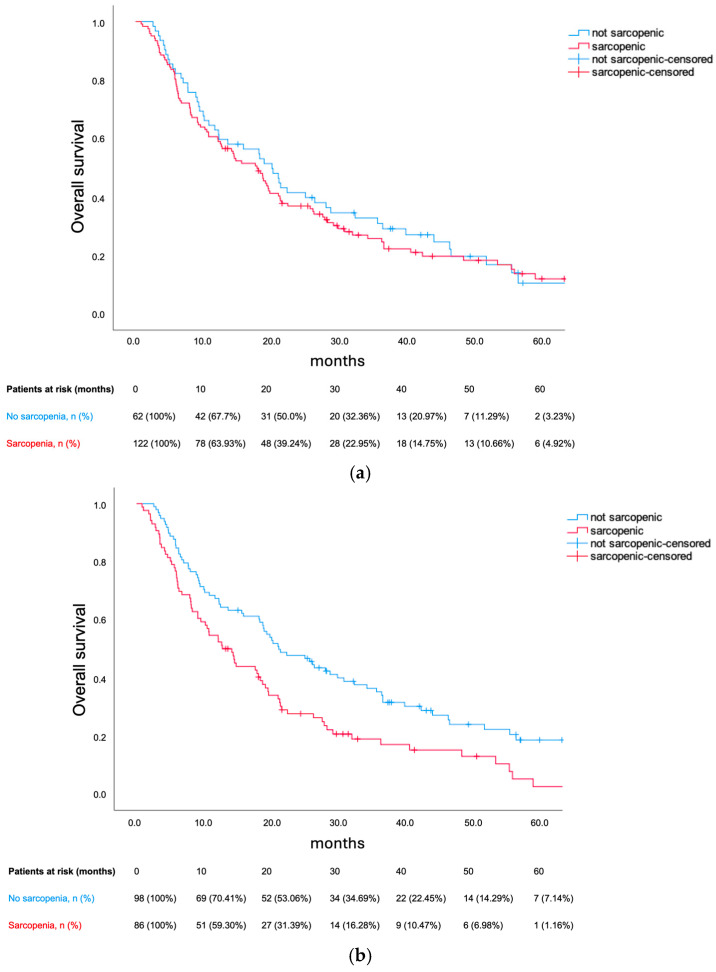
(**a**) Overall survival in the skeletal muscle index-assessed cohort with sarcopenia defined according to Prado et al. Median overall survival was 17.9 months for sarcopenic patients versus 20.11 months for not sarcopenic patients (*p* = 0.58). (**b**) Overall survival in the skeletal muscle index-assessed cohort with sarcopenia defined according to the cohort-specific thresholds. Median overall survival was 12.75 months for sarcopenic patients versus 21.13 months for not sarcopenic patients (hazard ratio for death 1.654; *p* = 0.002).

**Figure 2 cancers-17-00539-f002:**
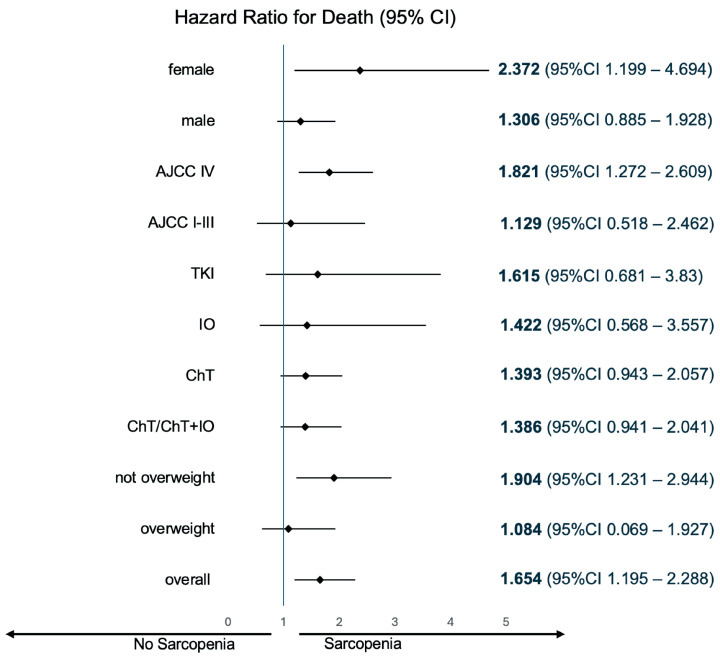
Forest plot for subgroup analysis of overall survival. CI, confidence interval; AJCC, American Joint Committee on Cancer; TKI, tyrosine kinase inhibitor; IO, immunotherapy; ChT, chemotherapy; and ChT/ChT + IO, chemotherapy or chemoimmunotherapy. Not overweight corresponds to a body mass index < 25 kg/m^2^, i.e., underweight or normal weight and overweight corresponds to a body mass index ≥ 25 kg/m^2^, i.e., overweight or obese.

**Figure 3 cancers-17-00539-f003:**
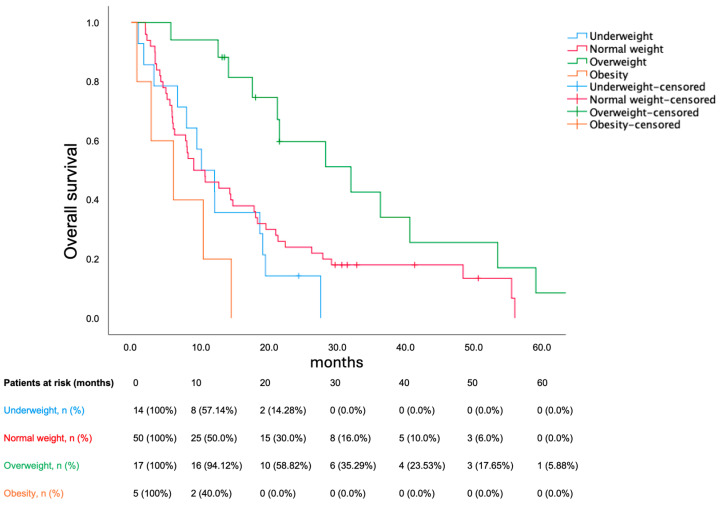
Overall survival in sarcopenic patients according to BMI. BMI, body mass index. Underweight corresponds to a BMI < 18.5 kg/m^2^; normal weight corresponds to a BMI ≥ 18.5 kg/m^2^ and <25 kg/m^2^; overweight corresponds to BMI ≥ 25 kg/m^2^ and <30 kg/m^2^; and obesity corresponds to BMI ≥ 30 kg/m^2^.

**Table 1 cancers-17-00539-t001:** Baseline characteristics of the initial cohort. ECOG, Eastern Cooperative Oncology Group; AJCC, American Joint Committee on Cancer; PD-L1, programmed death ligand 1.

Variable	Total (*n* = 197)
**Age**, mean ± standard deviation	65 ± 11.31
**Sex**, *n* (%)	
Male	135 (68.53%)
Female	62 (31.47%)
**Smoking status**, *n* (%)	
(Former) Smoker	103 (52.28%)
Never Smoker	30 (15.23%)
Unreported	64 (32.49%)
**ECOG performance status**, *n* (%)	
0	50 (25.38%)
1	106 (53.81%)
≥2	41 (20.81%)
**AJCC stage**, *n* (%)	
I–III	43 (21.83%)
IV	154 (78.17%)
**Histology**, *n* (%)	
Adenocarcinoma	165 (83.76%)
Squamous cell carcinoma	23 (11.68%)
Other	9 (4.57%)
**Metastatic sites**, *n* (%)	
≤2	156 (79.19%)
>2	41 (20.81%)
**PD-L1 tumor proportion score**, *n* (%)	
<1%	92 (46.70%)
1–50%	42 (21.32%)
>50%	40 (20.3%)
Unreported	23 (11.68%)
**First-line treatment**, *n* (%)	
Chemotherapy	113 (57.36%)
Immunotherapy	35 (17.77%)
Tyrosine kinase inhibitor	36 (18.27%)
Chemoimmunotherapy	13 (6.60%)

**Table 2 cancers-17-00539-t002:** Body composition data of the skeletal muscle index-assessed cohort. BMI, body mass index; SMI, skeletal muscle index. ^1^—<52.4 cm^2^/m^2^ for men and <38.5 cm^2^/m^2^ for women; ^2^—<49.96 cm^2^/m^2^ for men and <34.02 cm^2^/m^2^ for women.

Variable	Total (*n* = 184)
**BMI group**, *n* (%)	
<18.5 kg/m^2^	18 (9.78%)
≥18.5 kg/m^2^ and <25 kg/m^2^	94 (51.1%)
≥25 kg/m^2^ and <30 kg/m^2^	72 (39.13%)
≥30 kg/m^2^	20 (10.87%)
**BMI** (kg/m^2^), mean ± standard deviation	
Male (*n* = 125)	24.15 ± 4.75
Female (*n* = 59)	24.27 ± 4.12
**SMI** (cm^2^/m^2^), mean ± standard deviation	
Male (*n* = 125)	48.52 ± 9.31
Female (*n* = 59)	37.69 ± 6.14
**Sarcopenia (Prado et al.) ^1^,***n* (%)	
Male (*n* = 125)	89 (71.20%)
Female (*n* = 59)	33 (55.93%)
**Sarcopenia (cohort-specific) ^2^,***n* (%)	
Male (*n* = 125)	73 (58.40%)
Female (*n* = 59)	13 (22.03%)

**Table 3 cancers-17-00539-t003:** Univariate analysis for overall survival and multivariate analysis for overall survival, including statistically significant variables in univariate analysis. ECOG, Eastern Cooperative Oncology Group; PS, performance status; and CNS, central nervous.

Univariate Cox Regression Analysis			Multivariate Cox Regression Analysis	
Variable	*p*-Value	HR (95% CI)	*p*-Value	HR (95% CI)
Sarcopenia	0.002	1.65 (1.19–2.29)	0.019	1.50 (1.07–2.11)
Underweight	0.002	2.29 (1.37–3.86)	0.012	1.99 (1.16–3.40)
Overweight	0.074	-	-	-
Obesity	0.895	-	-	-
ECOG PS ≥ 1	0.009	1.68 (1.14–2.47)	0.008	1.69 (1.14–2.49)
Ab initio CNS M1	0.569	-	-	-
≥2 M1 sites	0.113	-	-	-
KRAS mutant	0.402	-	-	-
Squamous cell	0.581	-	-	-

## Data Availability

The original contributions presented in this study are included in the article. Further inquiries can be directed to the corresponding author.

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
