# Peer review of "Body Composition Analysis in Metastatic Non-Small-Cell Lung Cancer: Depicting Sarcopenia in Portuguese Tertiary Care"

_cancers, 2025, doi:10.3390/cancers17030539_

Round 1

Reviewer 1 Report

Comments and Suggestions for Authors

In this retrospective study, we aimed to evaluate the prognostic implications of using cohort-specific GMI thresholds versus literature-defined thresholds in Portuguese patients with metastatic non-small cell lung cancer (mNSCLC) receiving first-line palliative treatment. .

The study may be interesting, since it addresses a topic of great interest.

As suggestions, include this quote:  https://doi.org/10.3390/jcm13061601

On the other hand, you must indicate how you measure sarcopenia, that is, what precision you obtain with your measurement.

The sample is large, although there are more men than women. It would also be interesting to define the stage of diagnosis, since it influences survival, and analyze it as a variable.

On the other hand, I believe that the conclusions are not those of the study, that is, they seem like recommendations, I suggest that you write more specifically the results of your study in the conclusions, and remove the first person pronoun.

Reviewer 2 Report

Comments and Suggestions for Authors

Journal: CANCERS (ISSN 2072-6694)

Manuscript ID

cancers-3439250

Article Title:

Body composition analysis in metastatic non-small-cell lung cancer: depicting sarcopenia in Portuguese tertiary care

Dear Editor,

Thank you for inviting me to review this manuscript submitted to Cancers.

OVERALL COMMENTS

          Based on the statement that “Sarcopenia is an emergent prognostic biomarker in clinical oncology. Albeit increasingly defined through skeletal muscle index thresholding, literature cut-offs fail to discern heterogeneous baseline muscularity across populations.” The authors of this study (retrospective study) intended to assess the prognostic impact of using cohort-specific skeletal muscle index thresholds in a Portuguese metastatic non-small-cell lung cancer (mNSCLC) cohort. The overall survival (OS) was the primary endpoint, and the secondary included first-line (1L) progression-free survival and sarcopenia subgroup analysis regarding body mass index impact on OS. Their results showed that most tumors were adenocarcinomas with metastasis; 46.74% were sarcopenic patients. They concluded that reclassifying nearly one-fifth of the cohort's cohort-specific thresholds improved sarcopenia prognostication in mNSCLC.

TITLE

          The title is adequate.

_______

SIMPLE SUMMARY

          I suggest the simplification of the content of this section. It is made to help any reader understand it.

ABSTRACT

          This section is also adequate. However, I suggest removing the sentence “Conversely, Prado et al. definition lacked prognostic value” (line 64). It is not necessary here.

_______

KEYWORDS

          The included keywords were: Non-small-cell Lung Carcinoma; Sarcopenia; Body Composition; Prognosis; Biomarkers.

I suggest: Non-small-cell Lung Carcinoma; Sarcopenia; Body Composition; Skeletal Muscle Index; Prognosis.

_______

INTRODUCTION

This section is adequate. Please include more references published in the last 2 years.

METHODS

This section is also adequate. It only needs some modification, as suggested below.

On page 4, lines 136-137, we can read that “The study was done according to the Declaration of Helsinki and was approved by the Ethics Committee for Health of ULSSJ with a waiver for informed consent”. I suggest that the authors include the year of approval.

In lines, we can read 154-167: “The primary endpoint was overall survival (OS), defined as the time from mNSCLC diagnosis to death from any cause. The secondary endpoint 1L progression-free survival (PFS) was defined as time from starting 1L treatment in metastatic setting until disease progression or death from any cause. Two sarcopenia definitions – as published by Prado et al. (SMI 158 < 52.4 cm2 /m2 for men and < 38.5 cm2 /m2 for women) [14] and defined using SMI cohort-specific cut-offs) – were to be applied to statistical analysis based on relevance, i.e. both definitions were conditional to accurate survival stratification. Obesity was defined according to the World Health Organization definition (body mass index [BMI] ≥ 30 kg/m2 ). Sarcopenic Obesity was defined as simultaneous obesity (i.e. BMI ≥ 30 kg/m2 ) and sarcopenia (as published by Prado et al.14 or as defined within the study population).” Please see that Prado et al. are cited twice. In The first, the authors include [14]; in the other, number 14 is superscript.

RESULTS

          This section was well done. Please include all the definitions used in the tables in a legend.

          Figures 1a and 1b are too small.

_________

DISCUSSION

          This section is fine. The authors could, in an adequate way, discuss their results based on the existing literature.

          I also appreciate the inclusion of limitations on lines 345-355.

_________

CONCLUSION

This section needs to be expanded. I suggest expanding it and including future perspectives.

_________

REFERENCES

          Please see the suggestions that I included in the Introduction section.

Round 2

Reviewer 1 Report

Comments and Suggestions for Authors

The authors have made the suggested changes.